# Advances in Diagnosis, Pathological Mechanisms, Clinical Impact, and Future Therapeutic Perspectives in Tay–Sachs Disease

**DOI:** 10.3390/neurolint17070098

**Published:** 2025-06-25

**Authors:** María González-Sánchez, María Jesús Ramírez-Expósito, José Manuel Martínez-Martos

**Affiliations:** Experimental and Clinical Physiopathology Research Group CTS-1039, Department of Health Sciences, School of Health Sciences, University of Jaén, E23071 Jaén, Spain; mgs00080@red.ujaen.es (M.G.-S.); mramirez@ujaen.es (M.J.R.-E.)

**Keywords:** TSD, neurodegenerative disorder, hexosaminidase A, gemfibrozil, early death

## Abstract

Tay–Sachs disease (TSD) is a rare and severe neurodegenerative disorder inherited in an autosomal recessive manner. It is caused by a deficiency of the enzyme hexosaminidase A, which is responsible for the degradation of GM2 gangliosides—lipids that accumulate in the nerve cells of the central nervous system. The inability to break down these lipids leads to their progressive accumulation, resulting in irreversible brain damage. Mechanistically, TSD is caused by mutations in the *HEXA* gene, which encodes the alpha subunit of hexosaminidase A. These mutations disrupt enzyme activity and alter cellular pathways involved in lysosomal lipid degradation. Although Tay–Sachs specifically involves the alpha subunit, similar clinical features can be seen in Sandhoff disease, a related disorder caused by mutations in the *HEXB* gene, which encodes the beta subunit shared by hexosaminidase A and B. Tay–Sachs is classified into three clinical forms according to age of onset and symptom severity: the classic infantile form, which is the most common and severe; a juvenile (subacute) form; and an adult-onset form, which progresses more slowly and tends to present with milder symptoms. Diagnosis is based on enzymatic testing showing reduced or absent hexosaminidase A activity, confirmed by genetic testing. Prenatal diagnosis and genetic counseling play a key role in prevention and reproductive decision-making, especially in high-risk populations. Although no curative treatment currently exists, ongoing research is exploring gene therapy, enzyme replacement, and pharmacological approaches. Certain compounds, such as gemfibrozil, have shown potential to slow symptom progression. Early diagnosis and multidisciplinary care are essential to improving quality of life, although therapeutic options remain limited due to the progressive nature of the disease.

## 1. Introduction

The concept of rare diseases encompasses a wide range of distinct conditions that are often poorly understood due to limited scientific knowledge [1]. These disorders can manifest at any age and present with diverse clinical features, varying degrees of severity, and different patterns of progression [2]. The complex care required by individuals affected by rare diseases demands an integrative approach that considers not only the clinical aspects of the disease but also the impact on the patient’s family, social environment, and overall well-being [3]. The uncertainty surrounding diagnosis, prognosis, and treatment frequently contributes to psychological distress, including anxiety and depression [4].

A thorough understanding of the underlying pathological mechanisms has been key to developing therapies for genetic diseases, particularly in identifying optimal therapeutic windows. Nevertheless, many genetic conditions remain managed primarily through symptomatic treatment due to incomplete knowledge of disease pathogenesis and molecular targets. Recent advances in gene therapy have opened new possibilities for treating monogenic disorders, even in the absence of fully elucidated pathogenic mechanisms. However, effective interventions still depend on understanding the disease state, the timing of therapeutic intervention, and the extent to which disease progression is reversible.

In 1881, Dr. Warren Tay first described a severe, early-onset neurodegenerative disorder characterized by the presence of a “cherry-red spot” on the retina in infants of Jewish descent. In 1887, Bernard Sachs referred to this condition as “amaurotic familial idiocy”. By 1910, Sachs and Strauss had documented the accumulation of lipoid substances within ganglion cells, accompanied by distinctive balloon-like swelling of the dendrites [5]. The disorder was subsequently termed Tay–Sachs disease (TSD), and in 1970, it was determined to result from a deficiency of the isoform A of the enzyme hexosaminidase (HexA) [6]. In the late 1980s, the gene encoding the HexA enzyme was identified [7,8]. Around this time, it was also recognized that HexA deficiency could present in various forms, depending on the age of onset, leading to the classification of “late infantile,” “juvenile,” and “late onset” types [8,9].

TSD is a rare pediatric metabolic disorder and is classified among lysosomal storage diseases affecting the nervous system [10]. As a rare disease, it is cataloged in the ORPHANET database under reference number ORPHA845 [11] and in the OMIM database under reference number 272800 [12].

TSD is a fatal neurodegenerative disorder caused by a deficiency of the enzyme β-N-acetylhexosaminidase A (HexA), which is essential for the degradation of GM2 ganglioside within cells. HexA is a heterodimer composed of alpha and beta subunits. Mutations in the *HEXA* gene, which encodes the alpha subunit, result in TSD, whereas mutations in the *HEXB* gene, encoding the beta subunit, cause Sandhoff disease (SD). Clinically, these two disorders are often similar. Nevertheless, there are some distinguishing characteristics that are becoming more apparent through natural history studies [13].

The development of this disease is characterized by a wide range of symptoms and signs resulting from the progressive degeneration of the central nervous system [14]. GM2 gangliosidoses encompass three distinct disorders: TSD, SD, and the AB variant. SD differs from the other two by its systemic involvement, which can include hepatosplenomegaly, cardiomegaly, macroglossia, and skeletal abnormalities.

In a recent study, according to the initial clinical findings leading to the diagnosis, TSD was dominated by drowsiness, lethargy, and the child’s inability to sit independently. In Sandhoff’s disease, the cherry-red spot and hypotonia predominated [15].

Another current study investigated and demonstrated the existence of a late childhood GM2 gangliosidosis phenotype in late childhood TSD. Skeletal anomalies were found common in subjects with juvenile GM1 gangliosidosis, with kyphosis and hip dysplasia being the most prevalent. Furthermore, in subjects with GM1 and GM2 gangliosidosis, strabismus was a common finding [16].

GM2 ganglioside accumulation occurs throughout the body but is most prominent in the central nervous system, where gangliosides are particularly abundant. Consequently, both TSD and SD lead to progressive neurological deterioration. These disorders present with three recognizable clinical forms, which differ based on the age of onset and initial symptoms [15,16,17,18].

Infantile-onset, or type I TSD, is the most common and severe form. Children typically appear normal at birth but begin to show developmental regression around six months of age, accompanied by seizures and hyperacusis. Many patients with the infantile phenotype are found to have a cheery-red spot on the retina at time of diagnosis [15]. Death usually occurs by five years of age [19,20,21].

In contrast, juvenile-onset, or type II, manifests between two and ten years of age. Affected individuals often present with speech difficulties and clumsiness, followed by the development of spasticity and seizures. Cherry-red spots on the retina are often not present at the time of diagnosis, but may develop later as the disease progresses [16]. Optic atrophy tends to progress significantly. Death generally occurs during adolescence [22].

Finally, adult-onset, or type III, patients are frequently misdiagnosed for years. Clinical signs typically emerge during the second decade of life and may include ataxia, muscle weakness, and loss of ambulation. In the adult-onset Tay–Sachs disease, life span varies greatly. Some patients may have lifespan considerably reduced due to complications of the disease, while some patients live a nearly normal lifespan [21,22,23,24].

## 2. Epidemiology

TSD is rare in the general population, with an estimated incidence of approximately 1 in 100,000 live births in the United States and a carrier frequency of 1 in 250 individuals [25,26]. The disease is more prevalent in populations of Ashkenazi Jewish and French-Canadian ancestry, where its prevalence is significantly higher. For example, in the Ashkenazi Jewish population, the incidence rises to approximately 1 in 3900 live births without screening [25]. Epidemiological studies in the American Jewish community report a carrier frequency of 1 in 29 and a disease incidence of 1 in 3500 live births [27]. Despite this, due to pre-marital screening for carriers in many synagogues, the incidence of TSD in Ashkenazi individuals has been reduced by around 95% in some areas of the world, including the United States.

In contrast, the disease is sporadic in Asian populations, with an estimated prevalence of 1 in 360,000 live births [28]. To date, only two cases of TSD have been documented in Korea [28,29]. High-incidence clusters are also observed in specific communities, including the Old Order Amish in Pennsylvania, non-Jewish French-Canadians near the St. Lawrence River, and the Cajun population of Louisiana [30].

In high-risk populations, such as the Ashkenazi Jewish community, preventive strategies include preconception carrier screening programs aimed at informing reproductive decisions and reducing the likelihood of two carriers conceiving an affected child [31].

Over 220 distinct *HEXA* mutations associated with TSD are cataloged in mutation databases [13,32]. These mutations disrupt protein folding, intracellular transport, or structural stability, ultimately leading to loss of HexA enzymatic function [10,33].

## 3. Advances in Diagnosis

Therapy in GM2 disorders involves mitigation and reversal of GM2 ganglioside stored in pathology-relevant tissues. Minimizing the diagnostic odyssey through community and physician education and neonatal screening will be key to successful therapy, predominantly in rapidly progressive infantile-onset pathology. The diagnosis of patients with GM2 gangliosidosis is initiated by the recognition of clinical features of these disorders [34,35,36,37]. It may also be aided by neuroimaging characterized by basal ganglia hyperdensity which may also be the case for prominent but nonspecific cerebellar atrophy and other white matter changes [35,38,39].

A specific diagnosis requires the determination of HexA and HexB activities by using artificial substrates. However, the use of Hex can be analyzed by dried blood droplets, cells, tissues and biological fluids [40].

The gold standard for diagnosis is the measurement of enzyme activity in fibroblasts, chorionic villi or leukocytes [40] in addition to the inclusion of molecular diagnostics to determine the genotype of the patient. The β-hexosaminidase activity assay may not reliably identify carriers and therefore molecular diagnostics should be used for identifying carriers [35].

Heterogeneity of onset correlates inversely with residual Hex catabolic activity [41,42]. Patients with acute presentation have very low or absent enzyme activity, whereas patients with subacute or chronic onset have enzyme activity between 5 and 10% [43,44,45]. Notably, 10% wild-type enzyme activity prevents disease, with a base of studies describing degradation of GM2 ganglioside with activity between 10 and 15% [46].

### 3.1. Differential Diagnosis

#### 3.1.1. TSD Due to Activator Deficiency (AB Variant)

This variant presents with neuroregression, a cherry-red macular spot, absence of hepatosplenomegaly, and an exaggerated startle response. Unlike classic TSD, patients with the AB variant have normal levels of HexA and HexB. The accumulation of gangliosides is due to deficiency of the intralysosomal GM2 activator protein, which is necessary for the degradation of GM2 gangliosides [47].

#### 3.1.2. Sandhoff’s Disease and Other Lysosomal Storage Disorders

SD typically begins around 6 months of age and is characterized by progressive neurodegeneration, blindness, hyperacusis, and macular cherry-red spots. Clinically, SD is almost indistinguishable from TSD, except for the presence of visceral and skeletal involvement such as hepatomegaly, which is frequent in Sandhoff but not in Tay–Sachs. SD is caused by mutations in the HEXB gene and is not limited to any specific ethnic group. Other lysosomal storage disorders that may present with similar findings include GM1 gangliosidosis, infantile Gaucher disease, Niemann–Pick disease type A, and galactosialidosis [47].

#### 3.1.3. Differential Diagnosis of Late-Onset TSD

The differential diagnosis for late-onset TSD includes Friedreich’s ataxia, Kufs disease (adult-onset neuronal ceroid lipofuscinosis), amyotrophic lateral sclerosis, and late forms of other lysosomal storage disorders. Additional considerations include adolescent-onset spinal muscular atrophy, hepatolenticular degeneration, Niemann–Pick disease type C, cerebrotendinous xanthomatosis, metachromatic leukodystrophy, and X-linked adrenoleukodystrophy [47].

#### 3.1.4. Professionals Involved After Diagnosis

After diagnosing TSD, a multidisciplinary team is essential. Neurologists manage neurological symptoms, including electroencephalography, brain MRI, and antiepileptic treatment monitoring. Ophthalmologists assess visual impairment and its progression. Speech therapists evaluate swallowing and aspiration risk. Physiotherapists and occupational therapists address neuromuscular impairment. Psychologists are involved to evaluate and treat psychiatric symptoms, which may be the initial manifestation in up to half of patients. Psychiatric presentations can include classic or paranoid schizophrenia, often with bizarre behavior, agitation, disorganized speech, delusions, hallucinations, and severe personality changes [48]. Neurological features may be absent for years, while speech difficulties, cognitive impairment, and ataxia may occur, though frank dementia is uncommon. Speech impairment may also be associated with swallowing difficulties and risk of aspiration pneumonia.

## 4. Evaluation

The usual clinical findings of progressive weakness with inattention, developmental delay, exaggerated startle response, coupled with physical findings such as hyperreflexia, generalized hypotonia with clonus, or a cherry-red spot, warrant further evaluation for gangliosidosis. The first step in the evaluation is based on the measurement of serum levels of total hexosaminidase and hexosaminidase A. Individuals affected with the infantile form of this disease have enzyme activity of 0–5%, while those with the juvenile form have enzyme activity of 10–15% [49]. If the initial tests show reduced enzyme activity, molecular diagnostics should always be performed.

### 4.1. Genetic Studies

Molecular genetic testing utilizes duplication analysis, sequencing, and targeted analysis for pathogenic variants. Targeted analysis is indicated when enzyme activity is reduced or absent in the initial assay [50].

The standard panel includes six common pathogenic variants, three of which are null alleles associated with TSD in either homozygous or heterozygous states. The p.Gly269Ser variant is linked to adult-onset HexA deficiency. Two pseudodeficiency alleles (p.Arg247Trp and p.Arg249Trp) result in reduced degradation of synthetic substrates in vitro but do not cause neurological disease [40].

Interpretation of test results requires attention, as pseudodeficiency alleles do not reduce enzyme activity with the natural substrate in vivo [49]. Approximately 35% of non-Jewish individuals identified as heterozygotes in the HexA enzyme assay carry a pseudodeficiency allele, compared to 2% in the Jewish population.

Prenatal testing using fetal cells can be performed between 10 and 12 weeks (chorionic villus sampling) or 15 and 18 weeks (amniocentesis) of gestation in families where HexA enzyme analysis and molecular genetic testing exclude the presence of a pseudodeficiency allele in one parent and both parents are heterozygous [40].

### 4.2. Imaging Studies

Neuroradiological findings in TSD are described across three clinical phases.

In the initial phase, computed tomography (CT) reveals hypodensity of the basal ganglia and cerebral white matter, while magnetic resonance imaging (MRI) demonstrates hyperintense T2-weighted signal changes in these regions. The caudate nuclei are characteristically enlarged and protrude into the lateral ventricles during both the first and second phases. As the disease progresses, the extent of white matter hypodensity on CT increases, and in the final phase, there is marked cerebral atrophy [51,52,53,54].

MRI is superior to CT in delineating deep white matter demyelination. Involvement of the bilateral thalami is indicated by symmetrical T2-weighted hypointensity and T1-weighted hyperintensity, as well as hyperdensity on CT, findings that are suggestive of GM2 gangliosidosis, including TSD. Diffuse T2 hyperintensity and T1 hypointensity in the cerebral white matter reflect abnormal myelination due to ganglioside accumulation. In late stages, MRI shows advanced brain atrophy and diffuse white matter lesions, which are hyperintense on T2-weighted images [51,52,53,54].

Studies using magnetic resonance spectroscopy (MRS) in TSD have demonstrated increased choline: creatine ratio and increased myoinositol; creatine ratio and a decreased N-acetyl-aspartate; and creatine ratio, reflecting neuronal loss and gliosis. In late-onset or adult forms, pontocerebellar atrophy may be observed.

These imaging patterns, particularly the combination of bilateral thalamic involvement and progressive cerebral atrophy, are characteristic of TSD and aid in diagnosis and monitoring of disease progression.

## 5. Etiopathogenesis

### 5.1. Etiology of the Disease

TSD is part of a group of autosomal recessive lysosomal storage disorders known as GM2 gangliosidoses. These disorders are characterized by the accumulation of GM2 gangliosides within lysosomes, primarily affecting the nervous system and leading to neurodegeneration and neuronal dysfunction. The accumulation of GM2 gangliosides results from a deficiency of lysosomal β-hexosaminidase enzymes or the GM2 activator protein. Besides TSD, the GM2 gangliosidosis group includes SD and the AB variant (GM2 activator protein deficiency) [30].

TSD is specifically caused by mutations in the HEXA gene, which encodes the alpha subunit of the beta-hexosaminidase A enzyme located on chromosome 15q23. This enzyme is essential for the breakdown of GM2 gangliosides in neuronal lysosomes. To date, more than 130 mutations have been identified in the HEXA gene, including single base substitutions, insertions, duplications, splicing mutations, and complex gene rearrangements. These mutations impair the catalytic activity of the enzyme to varying degrees, resulting in a spectrum of clinical presentations, from classic infantile onset to later-onset forms [55].

The pathophysiology of TSD involves the storage of GM2 gangliosides in neurons, leading to neuronal loss, dendritic changes, and progressive neurological impairment, such as motor deficits, hypotonia, vision deterioration, and seizures. Diagnosis typically involves recognizing clinical features, measuring enzymatic activity, and confirming with genetic testing.

### 5.2. Pathogenesis of the Disease

The diagnostic criteria for TSD are based on the evaluation of neurological signs and symptoms, often using MRI and CT imaging to detect hypodensity in the basal ganglia and cerebral white matter [47]. Following neurological assessment, a blood test to measure HexA enzyme activity is performed; reduced or absent HexA activity confirms the diagnosis [47].

The clinical spectrum of TSD is divided into three main types based on symptom severity and age of onset. The acute infantile form is the most common and severe, with symptoms such as hypotonia, developmental regression, and seizures typically beginning around 6 months of age until death usually occurs by 3 years of age [15,56]. These patients have little or no functional HexA, leading to rapid GM2 accumulation [30].The juvenile subacute form begins between 3 and 5 years and is usually fatal by age 15 [22,30]. Common symptoms in the juvenile phenotype include ataxic gate and limb muscle weakness, along with gradually loss of speaking abilities over time [16,22]. This form is generally less aggressive than the infantile variant. The adult chronic form is defined by late onset of symptoms, which may not appear until adolescence or even the late twenties or early thirties [26,30]. Adult TSD is characterized by slower progression and a range of neurological and psychiatric symptoms.

## 6. Pathophysiology

Gangliosides are a group of glycosphingolipids primarily located in neuronal cell membranes, where they perform several biological functions essential for proper central nervous system function [42]. Approximately 5% of brain gangliosides correspond to GM2 gangliosides [57,58]. Under normal conditions, GM2 gangliosides are catabolized by lysosomal β-hexosaminidase hydrolases through hydrolysis of N-acetylgalactosamine residues [44]. Accumulation of GM2 gangliosides causes cytotoxic effects mainly in neurons, often leading to neuronal death [59].

Individuals with GM2 gangliosidosis exhibit progressive neurological deterioration, including hypotonia, progressive weakness, motor deficits, vision impairment, seizures, and decreased responsiveness [60]. Advances in understanding the pathophysiology and developing therapies for GM2 gangliosidosis have been supported by various animal models, including the SD cat, sheep with TSD, a TSD pig, SD mouse, and SD zebrafish, which replicate physiological and biochemical features of the disorder [61]. Recently, an asymptomatic murine model for TSD was developed that exhibits an alternative catabolic pathway for GM2 ganglioside [62]. Although several mutations in the feline *HEXB* gene have been described, no feline models for TSD currently exist [41,42]. A nonsense mutation in the *HEXA* cDNA causes TSD in Jacob sheep. Thus, the sheep model is the only animal that closely mirrors the disease progression, biochemical pathology, and genetics seen in human TSD. Due to similarities in central nervous system complexity and size relative to humans, evaluating therapeutic approaches in sheep is critical for translating findings to human medicine [63,64,65].

### 6.1. Structure and Physiological Function of Gangliosides

Gangliosides are complex glycolipids composed of a ceramide linked to a glycan containing at least one sialic acid residue [58,66]. More than 180 ganglioside species have been identified in vertebrates [58,66]. These molecules are primarily located in the outer leaflet of the plasma membrane, where they are concentrated in caveolae-rich microdomains or lipid rafts [66,67,68]. Within these domains, gangliosides perform essential functions, including the regulation of inflammation [57], neurite outgrowth [69,70], signal transduction [71], cell adhesion [72], membrane organization [67], and neuronal differentiation [67,69]. The structural diversity of gangliosides, determined by variations in their carbohydrate chains and ceramide moieties, underlies their ability to modulate a wide range of cellular processes and to participate in complex interactions with membrane proteins and other lipids.

### 6.2. β-Hexosaminidases: Synthesis, Transport and Catalytic Functions

β-Hexosaminidases are dimeric lysosomal enzymes composed of α and/or β subunits, forming three isoenzymes: HexA (αβ), HexB (ββ), and HexS (αα) [44]. The genes encoding the α (*HEXA*) and β (*HEXB*) subunits are located on chromosomes 15q23 and 5q13.3, respectively [44].

Post-translational modifications of these enzymes occur during ER-Golgi trafficking [73], after which the subunits dimerize to generate the active forms [73]. Dimerization of HexA and HexB is required for substrate degradation. HexA specifically hydrolyzes the N-acetylgalactosamine residue in GM2 ganglioside, while both HexA and HexB are capable of hydrolyzing glycosaminoglycans, glycoproteins, and glycolipids [44,73,74,75].

### 6.3. Clinical Presentations and Biochemical Correlates of GM2 Gangliosidosis

Although TSD, SD, and the AB variant are a consequence of mutations in distinct genes, the neurological involvement has many similarities in these disorders [42] (Figure 1). The typical neurological findings are associated with the timing of clinical onset. In the acute form, patients commonly present with hypotonia, regression of developmental milestones, and seizures [56]. The subacute form is characterized by intellectual disability, psychotic episodes, and motor regression [22]. In the chronic form, muscle weakness, cerebellar ataxia, dysphagia, and manic depression are frequently observed [60].

### 6.4. Pathophysiology of GM2 Gangliosidoses

Mutations in the GM2 activator protein or in the subunits of β-hexosaminidase result in the accumulation of GM2 gangliosides within lysosomes [42,44]. Early studies using animal models of SD demonstrated the presence of autoantibodies against GM2 gangliosides in both serum and the central nervous system [76], although the precise mechanisms remain unclear. This accumulation may promote lysosomal disruption and the subsequent release of gangliosides [77]. Since lysosomes are essential for the degradation of various macromolecules and organelles, their dysfunction disrupts cellular homeostasis and negatively impacts overall tissue physiology [78,79,80].

### 6.5. Neurodevelopmental Process

The use of SD brain organoids generated from patient-derived induced pluripotent stem cells (iPSCs) has provided valuable insights into the effects of GM2 ganglioside accumulation during neurodevelopment [81]. Recent studies have shown that impaired β-hexosaminidase activity in these organoids leads to increased organoid size and enhanced cellular proliferation, findings that were reversed by introducing HEXA and HEXB cDNAs using adeno-associated virus vectors [82]. Similarly, in HEXB-deficient zebrafish embryos, researchers observed an increase in lysosomal speckles in radial glia and, at five days post-fertilization [83], a reduction in lysosome content within microglia, although no increase in apoptosis was detected [82,83]. Collectively, these results demonstrate the cellular and functional consequences of GM2 ganglioside accumulation on nervous system maturation, highlighting the impact of this accumulation on early neurodevelopmental processes [84,85].

### 6.6. Neuronal Death and Neuroinflammation

Neuronal death is recognized as a central mechanism in the pathophysiology of GM2 gangliosidoses [44]. Postmortem analyses of brain and spinal cord tissue from TSD and SD patients have revealed elevated DNA fragmentation via in situ labeling, suggesting that apoptosis contributes to neurodegeneration [86]. Similar observations in TSD murine models [62] and SD animal models [87] demonstrated marked reductions in neuronal density [85]. While recent studies implicate GM2 ganglioside in triggering endoplasmic reticulum stress, the precise pathways linking this accumulation to increased neuronal apoptosis remain incompletely resolved [59,88]. Experimental evidence suggests that GM2 ganglioside induces neurite atrophy and cell death through PERK (protein kinase RNA-like ER kinase)-mediated apoptotic signaling [89].

Early apoptotic events, such as phosphatidylserine externalization on neuronal surfaces, correlate with characteristic microgliosis and immune cell infiltration observed in both patients and animal models [90,91]. These neuroinflammatory responses are thought to arise secondary to neuronal death [44,87,92]. Microglial proliferation and activation are well-documented features in SD and TSD models [92,93], while astrogliosis has been identified as a critical component of GM2 gangliosidosis pathophysiology, even during asymptomatic stages [94,95]. Emerging data suggest that astrocyte-microglia crosstalk is essential for amplifying neuroinflammatory responses, potentially driving neuronal injury through cytokine release and sustained ganglioside accumulation, thereby exacerbating neurodegeneration.

## 7. Histopathology

TSD has been identified in all major brain regions, including the cervical and lumbar spinal cord. Tissue samples from multiple animals at three, six, and nine months of age, as well as at the humane endpoint, were evaluated using light microscopy with histochemical and immunohistochemical staining. The cervical and lumbar spinal cord, substantia nigra, hippocampus, cortex, and thalamus were available from at least two animals per group, while the cerebellum was represented by one animal per group. The red and oculomotor nuclei were examined in samples from at least two time points. The disease in these animals was progressive [96].

A notable finding was the consistent presence of astrocytic changes as early as three months, with the most severe alterations observed between six and nine months. These reactive astrocytes were characterized by increased cytoplasmic prominence, particularly in the limiting glia and subcortical white matter. There was also a mild to moderate increase in the number of visible microglial nuclei [44].

Additionally, scattered vacuolization of the neuroparenchyma and the formation of rare small spheroids were observed, with these changes progressing over time and being most pronounced in the telencephalic white matter. In the cerebellum, Purkinje cell bodies were enlarged and contained storage material at all time points, accompanied by increased cytoplasm in granular neurons and a decrease in the cellularity of the granular layer. These findings are consistent with the widespread neurodegeneration and glial response characteristic of TSD [44,97,98].

### 7.1. Neuroinflammation

The neuroinflammatory response in sheep with TSD is characterized by increased immunohistochemical labeling for astrocytes (using glial fibrillary acidic protein, GFAP) and microglia. In the parietal cortex and cerebellar hemispheres, there is a notable increase in GFAP labeling as the disease progresses, especially by 9 months of age, indicating pronounced astrogliosis in these regions. Interestingly, scientific evidence does not show an increase in GFAP in the hippocampus, which may be related to the differential expression of ionotropic glutamate receptors by astrocytes in the cortex but not in the hippocampus [99]. Additionally, glial densities in the thalamus and parietal cortex are elevated early in the life of affected animals, reflecting a widespread activation of glial cells as part of the neuroinflammatory process.

A study by Jarnes-Utz et al. evaluated 188 biomarkers in the CSF and serum of patients with infantile and juvenile gangliosidosis, and found 5 biomarkers of inflammation were consistently elevated in the CSF of patients with the infantile phenotype. These biomarkers were epithelial-derived neutrophil-activating protein 78 (ENA-78), monocyte chemotactic protein 1 (MCP-1), macrophage inflammatory protein-1 alpha (MIP-1α), macrophage inflammatory protein-1 beta (MIP-1β), and tumor necrosis factor receptor 2 (TNFR2) [100].

### 7.2. Neurodegeneration

Microtubule-associated protein 2 (MAP2) stabilizes neuronal microtubules and is used as an indicator of neuronal loss or dendritic process area in sheep with TSD. Microscopic evaluation shows that the area of MAP2 labeling in the thalamus at three months of age is equivalent to that of normal controls [101].

## 8. Clinical Manifestations

TSD is classified into infantile, juvenile, and adult forms based on the age of onset. Early diagnosis is challenging due to nonspecific biochemical findings and clinical features [102].

### 8.1. Childhood TSD

Infantile TSD is a prototypical neurodegenerative disorder affecting the gray matter during infancy. Affected infants are typically born without apparent abnormalities. Although symptoms may rarely appear in the first week of life, onset most commonly occurs between 3 and 6 months of age. In rare cases, unassociated hydrops fetalis, characterized by abnormal fluid accumulation in at least two fetal compartments, may represent a prenatal manifestation [103].

Initial clinical features include irritability, mild motor weakness, and hypersensitivity to auditory and sensory stimuli. An exaggerated startle response is an early and useful diagnostic sign. The presence of a cherry-red spot on the retina, observed during fundus examination, is considered highly specific for TSD [21,104,105]. Cherry-red spots are not always present at the time of diagnosis. It is important to note that cherry-red spots may develop later if they are not present at time of diagnosis. This finding results from pallor caused by swollen retinal ganglion cells, which accentuates the underlying choroidal vasculature. The cherry-red spot is typically present by 6 months of age, with vision loss developing between 12 and 18 months; by 30 months, most patients are blind. Additional ophthalmologic findings include nystagmus, optic atrophy, and narrowing of retinal vessels [106].

Neurological symptoms are the hallmark of the disease. Between 4 and 6 months, infants exhibit developmental regression and delays. Symptoms progress rapidly between 8 and 10 months, with a marked reduction in spontaneous and voluntary movements and increasing unresponsiveness [56,107]. By 12 months, tonic–myoclonic seizures develop, and spasticity and refractory seizures characterize the later stages. More than two-thirds of patients require multiple anticonvulsants for seizure control [108]. Various seizure types, including gelastic, focal, and generalized, may occur. Around this time, affected individuals may also develop dyskinesia, sleep disturbances, ataxia, episodes of screaming, and irritability. By 18 months, macrocephaly, due to reactive cerebral gliosis rather than hydrocephalus, is commonly observed [109]. At 2 years of age, cognitive impairment, decerebrate posturing, dysphagia, and progression to a vegetative state are typical.

Cardiovascular complications are rare but may result from substrate accumulation, with reports of prolonged QT intervals and nonspecific T-wave changes [110]. Hepatosplenomegaly is generally absent. Patients are prone to infections, particularly respiratory, which are a leading cause of death. Carriers of TSD demonstrate increased resistance to mycobacterial infections, attributed to elevated levels of the HexB beta subunit [111].

### 8.2. Juvenile TSD

Juvenile TSD presents between 2 and 10 years of age and is caused by reduced activity of the HexA enzyme. Earlier symptom onset is associated with a more rapid disease progression. Initial manifestations commonly include clumsiness, muscle weakness, and incoordination. As the disease progresses, spasticity, dysphagia, dysarthria, and ataxia become progressively more severe [16,61]. The presence of a cherry-red spot on fundoscopic examination is inconsistent in this form. In later stages, optic atrophy and retinitis pigmentosa may develop [16,112]. By 10 to 15 years of age, affected individuals typically enter a vegetative state characterized by decerebrate posturing, with death occurring a few years later, most often due to respiratory infections.

### 8.3. TSD in Adults

Adult onset TSD (type III) progresses more slowly than the infantile and juvenile forms, and this is thought to be due to higher residual HEXA enzyme activity, which commonly ranges between 5% and 20% of normal levels. Symptom onset often occurs in adolescence or early adulthood, may also occur in the 3rd to 4th decade in life, or later [113].

Clinical manifestations are heterogeneous and may include cognitive impairment, psychiatric symptoms such as depression, and cerebellar signs including dysarthria, dystonia, ataxia, and tremor. Psychiatric features are common, with approximately 30–50% of individuals developing psychiatric manifestations without dementia, such as psychotic depression, paranoia, hallucinations, and bipolar symptoms. In some cases, psychiatric symptoms may precede neurological findings, and the presence of cognitive decline or neurological symptoms in individuals with psychiatric illness should raise suspicion for storage disorders [114,115].

Progression is generally slow, and affected individuals may also develop muscle weakness, clumsiness, loss of coordination, tremors, and difficulties with speech and swallowing. Over time, some patients experience executive dysfunction and memory impairment. The clinical course and severity are highly variable, even among affected family members [114,115].

## 9. Forecast

TSD is a progressive neurodegenerative disorder. In the infantile form, neurological deterioration advances rapidly, and seizures are often refractory to treatment. Even with optimal professional care, affected children typically die by 4 to 5 years of age, most often due to recurrent infections such as pneumonia [11,47]. In late-onset forms, patients experience progressive motor impairment and gait disturbances, often requiring adaptive equipment and mobility assistance. Psychiatric symptoms are frequent and frequently resistant to treatment. Progressive neurological decline in juvenile forms usually leads to a vegetative state and death between 10 and 15 years of age [47]. The late-onset (adult) form progresses more slowly; while it causes significant motor and psychiatric symptoms, it does not always shorten life expectancy.

## 10. TSD Models

### 10.1. In Vitro and In Vivo Models of TSD for Therapy Development

To evaluate the efficacy of novel therapies, in vitro models using induced pluripotent stem cells derived from fibroblasts of patients with childhood-onset TSD have been proposed [116]. These lines were much more distinct in the central nervous system, exhibiting a phenotype of lysosome and lipid accumulation pathology. Treatment of these cells with human HexA proteins reduced lipid and lysosome accumulation [117,118]. Human recombinant lysosomal beta-hexosaminidases produced in *Pichia pastoris* efficiently reduced lipid accumulation in Tay–Sachs fibroblasts [75].

The initial in vivo models included mice and sheep. The primary mouse model for TSD was established in 1995 through targeted knockout of the *HEXA* gene, resulting in mice lacking HexA activity [119]. Despite this deficiency, these mice exhibited accumulation of GM2 gangliosides and formation of membrane cytoplasmic bodies only in select brain regions, excluding the forebrain, olfactory bulb, and cerebral cortex. Notably, these mice did not develop clinical symptoms of TSD and maintained a normal lifespan. In contrast, mice deficient in *HEXB* develop central nervous system neurodegeneration, muscle weakness, tremor, ataxia, and spasticity [120,121], distinguishing them from *HEXA*-deficient mice and making them useful for initial evaluation of potential treatments for GM2 gangliosidosis. It has also been described that abnormally accumulated GM2 gangliosides contribute to skeletal deformity in TSD mice. Hexa-/-Neu3-/- mice exhibit kyphosis in the late progressive stage of the disease [62]. Mouse models with a deficiency linking the *Hexa* and *Neu3* genes showed different anomalies in both the number and size of lysosomes, predominantly in the brain, due to the abnormal accumulation of GM2. Specifically, Hexa-/-Neu3-/ mice have neurodegeneration that increases with neuronal loss and depletion of Purkinje cells, surviving up to 5 months [62,122]. Furthermore, the abnormal increase in lipids in TSD leads to a disruption in the fusion between autophagosomes and lysosomes, causing an increase in autophagosomes [62,122,123].

CRISPR/Cas9-based genome editing using Cas9 nickase (D10A), resulted in a significant effect on the target and no detectable off-target effect. Specifically, this study evaluates the editing of this gene to insert normal HEXA and HEXB cDNA into the AAVS1 locus in in vitro models of TSD and SD [76,124,125].

TSD has also been documented in other animal species, such as the American flamingo (*Phoenicopterus ruber*) [126] and Jacob sheep [127]. In these animals, the disease arises spontaneously and is characterized by GM2 ganglioside accumulation and HexA deficiency [61]. Among these models, the clinical manifestations observed in Jacob sheep most closely resemble those seen in humans, making this species the most relevant large animal model for studying TSD and evaluating therapeutic strategies [127].

Regardless of the therapeutic modality under development, the identification of effective probes relies on reproducible and robust in vitro assays and in vivo models that accurately represent the human disease. In vitro assays for TSD are typically divided into two main categories: phenotypic assays and HexA enzyme activity assays. The choice of assay readout depends on the mechanism of action of the therapeutic being tested. Therapeutic approaches aimed at correcting HexA enzyme production, activity, or trafficking commonly utilize HexA enzyme activity assays in cell lysates to assess efficacy [128].

Available in vivo models for TSD include sheep, mice, and rabbits. The primary murine model was developed in 1995 by Taniike et al. [119] through targeted knockout of the HEXA gene. Among animal models, sheep are notable for displaying clinical features that closely resemble those observed in human TSD, making them particularly valuable for translational research [26,64,65,93,127,129].

### 10.2. HexA Enzyme Activity Assays

Synthetic fluorogenic substrates are used to assay the Hex activity of cell lysates or purified enzymes. Total Hex activity is measured by 4-methylumbelliferyl-β-N-acetylglucosamine (MUG), whereas 4-methylumbelliferyl-β-N-acetylglucosamine-6-sulfate (MUGS) is used to measure HexA-specific activity [130].

## 11. Treatment

### 11.1. Non-Pharmacological Treatment

Currently, no effective treatment for GM2-gangliosidosis has been found, although several clinical trials are currently underway [11]. Management of TSD is primarily supportive, focusing on airway protection, management of infectious complications, seizure control, adequate nutrition, early physiotherapy, and intensive occupational therapy.

Various strategies have been explored to develop targeted therapies for GM2 gangliosidosis, including enzyme replacement therapy, biotechnological approaches, and gene editing, with studies conducted both in vitro and in vivo. However, no treatment has been approved for Tay–Sachs or SD to date, and there is currently no effective or curative therapy available. Nursing care plays a critical role in the management of Tay–Sachs patients. The development of individualized care plans is essential, as no standardized plan currently exists. These care plans should be tailored to the specific needs of each patient and be based on the most current scientific knowledge rather than generic protocols.

### 11.2. Enzyme Replacement Therapy

Enzyme replacement therapy (ERT), first conceptualized by Christian de Duve in 1964, is based on the administration of lysosomal enzymes that are taken up by endocytosis and delivered to lysosomes [131]. ERT has been approved for several lysosomal storage disorders, including Gaucher, Pompe, Fabry diseases, and alpha-mannosidosis [132]. The first ERT for Gaucher disease was approved by the U.S. Food and Drug Administration in 1991, utilizing glucocerebrosidase purified from human placenta, later replaced by the recombinant enzyme imiglucerase [133]. Currently, recombinant DNA technology is used to produce ERTs. These proteins enter lysosomes via endocytosis, where they exert their function to improve the phenotype of the affected individuals, and are administered intravenously. However, the efficacy of ERT varies among patients, and not all tissues and organs respond equally to this therapy [134].

Importantly, these enzymes cannot cross the blood–brain barrier (BBB), limiting their therapeutic effect on neurological symptoms. In GM2 gangliosidosis, early studies by Johnson et al. [135] using intravenous HexA in a patient with SD, showed enzyme activity in the liver but not in the cerebrospinal fluid or brain parenchyma, indicating the enzyme does not cross the BBB [136]. To address this, fusion proteins known as molecular Trojan horses have been developed. These are monoclonal antibodies that recognize the transferrin or human insulin receptor, allowing the enzyme to cross the BBB via receptor-mediated endocytosis.

Alternative administration routes, such as direct injection into the cerebrospinal fluid or intrathecal injections, have shown potential therapeutic effects [131,137,138]. For example, intrathecal administration of HexA produced in Chinese hamster ovary cells or yeast *Ogataea minuta* has been tested in SD mouse models [44,139,140]. ERT has also been shown to reduce GM2 ganglioside levels in the liver, which is significant since hepatosplenomegaly is a frequent finding in SD [141]. Enzymes used in these therapies are produced in CHO cells, and recombinant Hex produced in *O. minuta* and *Pichia pastoris* has also been evaluated [140,142].

For TSD, ERT is particularly challenging because effective therapy requires both functional alpha and beta subunits of HexA. Several groups have developed HexA enzymes that alleviate pathological symptoms in vitro [140,143]. However, due to the large size of HexA, it cannot cross the BBB, and intravenous administration is not effective for central nervous system symptoms [26].

### 11.3. Hematopoietic Stem Cell Transplantation

Hematopoietic stem cell transplantation (HSCT) is primarily performed using bone marrow, umbilical cord blood, or peripheral blood stem cells [30,76]. This approach is based on the ability of Hex enzymes to transfer between cells via the mannose-6-phosphate receptor (M6PR), allowing for export to the extracellular space [76,144,145] and cross-correction of neighboring cells through an M6PR-mediated mechanism [44,146], resulting in sufficient enzyme levels either naturally or through engineering [146,147].

Only a small number of patients with a diagnosis of GM2-gangliosidosis have undergone HSCT. Jacobs et al. [148] reported the use of allogeneic bone marrow transplantation in an asymptomatic 3-year-old with subacute TSD, which increased HexA levels but did not prevent neurodegeneration. In another series, five children with childhood-onset TSD underwent transplantation for unrelated ulcerative colitis; survival was prolonged in two cases, with arrest of neurodegeneration but no improvement in motor skills [149]. Clinical studies and animal models indicate that while HSCT can normalize or reduce circulating substrates such as GM2, it does not provide sufficient enzymatic activity in the central nervous system to halt disease progression or reverse neurological symptoms. Significant challenges for HSCT include finding a compatible donor and managing immunogenic responses from imperfect matches.

Recent research has focused on gene therapy-corrected autologous stem cells or transplantation of healthy donor stem cells [76]. The fundamental properties of stem cells, such as multipotency, self-renewal, and proliferation, remain intact, supporting the potential of ex vivo gene therapy as an ideal approach for GM2 gangliosidosis [150]. Autologous stem cell transplantation, in which gene therapy is used to correct mutations in hematopoietic precursors isolated from patients, is a promising alternative [147].

### 11.4. Pharmacological Chaperones

Pharmacological chaperones are small molecules that bind specifically to target proteins, promoting correct folding and stabilization [151,152]. These compounds act in the endoplasmic reticulum, binding with high affinity to the misfolded protein and facilitating its proper folding and trafficking to the lysosome. Upon reaching the lysosome, the acidic pH and presence of natural substrate promote dissociation of the chaperone from the enzyme, allowing normal enzymatic function [153,154].

Most pharmacological chaperones act as competitive inhibitors, binding to the active site of the enzyme and stabilizing its structure, although non-inhibitory chaperones targeting allosteric sites have also been identified [154,155]. The efficacy of pharmacological chaperones is mutation-dependent, limiting their use to patients with specific genotypes [153,154]. It is important to distinguish pharmacological chaperones from chemical chaperones, such as dimethyl sulfoxide, which are less selective and often associated with higher toxicity [156].

Pyrimethamine has been identified as a promising pharmacological chaperone for HexA, inducing up to a threefold increase in enzymatic activity in TSD fibroblasts [157,158]. Pyrimethamine, a drug used for malaria and cerebral toxoplasmosis, can cross the blood–brain barrier and has been shown in clinical trials to increase HexA activity. However, the effect on neurological symptoms is limited, and adverse effects have been reported [44,159,160].

Additionally, progranulin, a glycoprotein secreted by epithelial, immune, and neuronal cells, has been shown to enhance the folding of mutant Hex, and is involved in various physiological and pathological processes, including neurodegeneration [161].

### 11.5. Substrate Reduction Therapy

Substrate reduction therapy (SRT) is a therapeutic approach that aims to partially inhibit the synthesis of substrates that accumulate due to enzyme deficiencies in lysosomal storage diseases, thereby reducing the burden on the defective enzyme and decreasing substrate accumulation. In TSD, SRT targets the reduction in GM2 ganglioside production, the substrate that accumulates as a result of HEXA deficiency.

Miglustat (N-butyldeoxynojirimycin, NB-DNJ) is a small molecule competitive inhibitor of glucosylceramide synthase, the enzyme responsible for the first committed step in glycosphingolipid synthesis. Miglustat has demonstrated efficacy in murine models of Tay–Sachs and SD [162], reducing GM2 ganglioside accumulation in the brain by up to 50% and prolonging survival. In clinical studies, miglustat has been administered to patients with juvenile GM2 gangliosidosis and chronic SD. In a 24-month study of five patients with juvenile GM2 gangliosidosis, miglustat did not halt neurological deterioration, although some neuropsychological tests showed stability in certain cognitive functions, and brain MRI findings did not worsen in most patients [163]. In a three-year follow-up of a patient with chronic SD, miglustat produced only minor effects on neurological progression [164].

Despite these findings, miglustat has not demonstrated significant clinical benefit in preventing or reversing neurological decline in GM2 gangliosidosis, and its efficacy appears limited, particularly in advanced disease stages. As a result, the potential benefit of miglustat or other SRTs may depend on early intervention, but further studies are needed to clarify their role. Miglustat has not been approved by the U.S. Food and Drug Administration for the treatment of Tay–Sachs or SD. Another SRT, Genz-529468, functions as a glucosylceramide synthase inhibitor similar to miglustat; however, its full mechanism of action and clinical efficacy are still under investigation [165].

### 11.6. Gene Therapy

Gene therapy is based on the use of viral vectors to deliver a functional gene to correct a genetic defect. Since TSD is a monogenic disorder, gene therapy represents a promising treatment approach for this pathology. Gene therapy can be administered ex vivo, where autologous cells are isolated from the patient, transduced with viral vectors, and then reintroduced into the body, or in vivo, where the gene therapy is delivered directly to the patient. Adeno-associated virus (AAV) is the primary vector used in in vivo gene therapy, while ex vivo approaches mainly utilize retroviral vectors such as gamma-retrovirus-derived lentivirus vectors or HIV-1. AAV vectors have been used in 250–300 clinical trials and have shown a favorable safety profile in humans [166,167,168].

Several gene therapies have been approved by the European Medicines Agency (EMA), including Libmeldy, which is authorized for ex vivo gene therapy in metachromatic leukodystrophy and is currently in clinical trials in the USA [169]. The ex vivo gene therapy strategy for lysosomal storage disorders involves transducing isolated hematopoietic stem cells from the patient with the gene of interest, using autologous cells as modified hematopoietic stem cell transfer therapy.

For in vivo gene therapy, since AAV does not cross the blood–brain barrier, clinical trials have utilized intrathecal or intracerebral administration in lysosomal storage disorders with neurological involvement, demonstrating good tolerance to gene therapies [170,171,172]. This is particularly promising for TSD due to its primarily neurological manifestations. Gene therapy remains an active area of development for lysosomal storage disorders, with several clinical trials ongoing and extensive preclinical research [172].

A unique challenge in gene therapy for TSD is that HEXA encodes a subunit of the heterodimeric enzyme HexA, which must be complex with the beta subunit encoded by HEXB to be functionally active. Therefore, gene therapy for TSD requires delivery of both subunits to produce a functional enzyme. One advantage of this strategy is the potential to treat both Tay–Sachs and SDs with a single therapy, although not the AB variant.

Notably, AAV-based therapies have successfully reduced GM2 ganglioside levels in cats and mice, however they have not decreased GM2 levels in the central nervous system of non-human primates [76,173,174].

Recent clinical studies have shown that AAV gene therapy using a combination of AAVrh8-HEXA and AAVrh8-HEXB vectors is feasible and well-tolerated in patients with infantile TSD. In these studies, patients received intrathecal or combined thalamic and intrathecal administration, resulting in stable increases in cerebrospinal fluid HexA activity and some evidence of disease stabilization, with no vector-related adverse events reported [175,176].

### 11.7. Ex Vivo Gene Therapy

Another innovative approach involves the transplantation of ex vivo modified multipotent neural cells expressing human HEXA, generated through retroviral transduction. This strategy aims to achieve sufficient distribution and production of HexA to exert a therapeutic effect in TSD [30,47]. Studies have shown that multipotent neural cells engineered to overexpress the human HEXA gene can stably secrete active HexA enzyme, cross-correcting the metabolic defect in Tay–Sachs patient-derived fibroblasts in vitro. Intracranial transplantation of these cells into mice resulted in widespread expression of the human HexA subunit and production of enzymatically active HexA throughout the brain at levels considered therapeutic for TSD [30,47].

Currently, enzyme replacement therapy, bone marrow transplantation, and substrate reduction therapy have demonstrated limited efficacy in preventing neurodegeneration in TSD. These approaches have not been able to halt disease progression or reverse neurological symptoms, especially when initiated after the onset of significant pathology. As myelination defects appear early and worsen over time, it is increasingly recognized that combining multiple therapeutic strategies at an early age may be necessary to achieve meaningful clinical benefit. Early intervention is critical, as delayed treatment may not adequately address early myelination defects or prevent irreversible neurological damage.

### 11.8. Pharmacological Treatment

Glial dysfunction and activation are hallmarks of neurodegenerative and neuroinflammatory diseases such as multiple sclerosis, Parkinson’s disease, and Alzheimer’s disease [122,177,178,179]. Recent studies have shown that central nervous system storage of GM1 and GM2 gangliosides leads to microgliosis and astrogliosis, with the degree of inflammation correlating with higher levels of ganglioside accumulation [122]. Consequently, agents capable of inhibiting inflammation and glial activation in Tay–Sachs pathogenesis may provide neuroprotection in TSD.

Gemfibrozil, a lipid-lowering drug, inhibits the expression of pro-inflammatory cytokines and inducible nitric oxide synthase in astrocytes and microglia [180,181,182]. It should be noted that this drug has only been investigated in murine models, not in humans [183]. Oral administration of gemfibrozil has been shown to decrease glial inflammation and glycoconjugate accumulation in the motor cortex of Tay–Sachs mice. In animal models, treatment with gemfibrozil reduced the amount of activated astroglia in Tay–Sachs mice, indicating an anti-inflammatory effect. Additionally, oral gemfibrozil decreased microglial inflammation.

The neuroprotective effect of gemfibrozil treatment against Tay–Sachs-associated brain pathology was evaluated by assessing behavioral parameters in experimental mice, focusing on the motor cortex, movement, and regions regulating motor coordination. Prior to treatment, Tay–Sachs mice exhibited increased toe separation and stride length. After oral administration of gemfibrozil, improvements in general motor ability were observed, with stride length and toe separation approaching normal values.

The effects of a ketogenic diet (KD) and propagermanium (PG), alone and in combination, have also been evaluated on chronic neuroinflammation in the Hexa-/-Neu3-/- mouse model of TSD [184]. PG, also known as 3-oxygermylpropionic acid polymer, is an organic germanium compound which blocks the neuroinflammatory response induced by Ccl2, which is highly expressed in astrocytes and microglia. Results showed that KD, PG, and the combined KD+PG treatment significantly reduced the gene expression levels of pro-inflammatory chemokines (Ccl2, Ccl3, Ccl5, Cxcl10) and the astrocyte marker GFAP in both the cortex and cerebellum. KD alone and the combined KD+PG treatments also decreased astrocyte activation in both regions, and KD alone significantly reduced macrophage/monocyte activity (MOMA-2) in the cortex and cerebellum, with the combination having this effect only in the cerebellum. Although KD effectively induced ketosis, none of the treatments altered GM2 ganglioside accumulation, improved neuromotor coordination, reduced anxiety-related behavior, prevented disease progression, or extended lifespan [184]. These findings suggest that modulating pro-inflammatory chemokines or their receptors may be prospective therapeutic targets to delay TSD neuropathology. The data supports the potential for implementing combined treatment strategies to reduce chronic inflammation in TSD and potentially other lysosomal storage diseases.

## 12. Conclusions

TSD is a rare, progressive neurodegenerative disorder caused by a deficiency of the enzyme ß-hexosaminidase A, leading to toxic accumulation of GM2 gangliosides in neurons and resulting in central nervous system dysfunction and neurodegeneration. The disease is inherited in an autosomal recessive manner and presents a diagnostic challenge due to its rarity, nonspecific biochemical findings, and variable clinical features, which require comprehensive neurological evaluation and specialized testing for accurate diagnosis. Diagnosis is based on clinical assessment, neuroimaging, and confirmation through enzyme activity assays and molecular genetic testing for mutations in the HEXA gene. Early and precise diagnosis is essential for management and genetic counseling.

Research into the pathophysiology of TSD has revealed complex mechanisms involving lysosomal dysfunction, impaired autophagy, and progressive neurodegeneration, but the exact link between GM2 accumulation and neuronal death remains incompletely understood. Multiple experimental therapies are under investigation, including enzyme replacement, gene therapy, and bone marrow transplantation, but currently, there is no disease-modifying treatment approved for clinical use.

Management remains supportive, focusing on airway protection, infection management, seizure control, nutritional support, physiotherapy, and occupational therapy. The prognosis for infantile-onset TSD is poor, with most affected children dying by 4 to 5 years of age despite optimal care, often due to recurrent infections. Juvenile and adult forms progress more slowly but still lead to significant neurological and psychiatric impairment. Continued translational research and the development of innovative therapies are essential to address the underlying causes of TSD and to improve outcomes for affected individuals.

## Figures and Tables

**Figure 1 neurolint-17-00098-f001:**
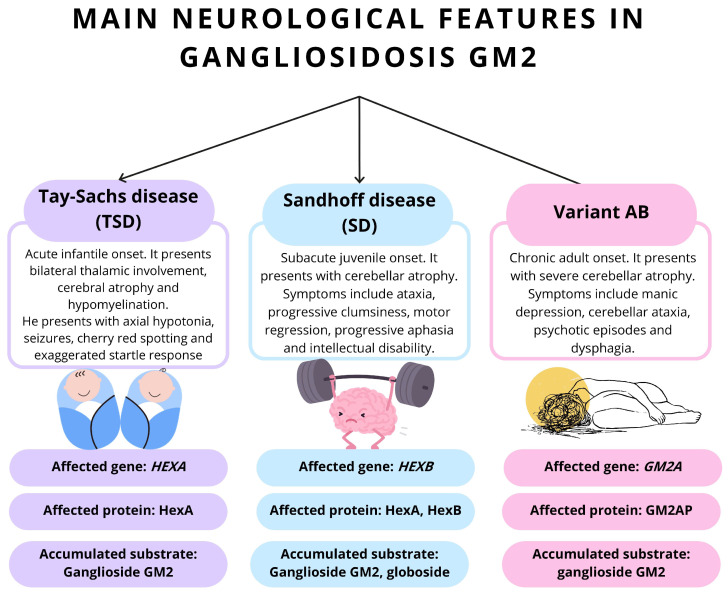
The main neurological features and molecular characteristics of GM2 gangliosidosis variants. The figure summarizes the clinical presentations, genetic defects, affected proteins, and accumulated substrates in the three principal forms of GM2 gangliosidosis: TSD, SD, and Variant AB. TSD, caused by mutations in *HEXA*, presents with acute infantile onset and is characterized by bilateral thalamic involvement, cerebral atrophy, hypomyelination, axial hypotonia, seizures, cherry-red macular spots, and exaggerated startle response. SD, resulting from *HEXB* mutations, typically manifests as a subacute juvenile-onset disorder with cerebellar atrophy, ataxia, clumsiness, motor regression, progressive aphasia and intellectual disability. Variant AB, due to *GM2A* mutations, is distinguished by chronic adult onset, severe cerebellar atrophy, manic depression, cerebellar ataxia, psychotic episodes, and dysphagia. The respective gene mutations lead to deficiency of HexA (TSD), HexA and HexB (SD), or GM2 activator protein (Variant AB), resulting in the accumulation of GM2 ganglioside in neuronal tissues. TSD is not typically considered gender specific.

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
