# Peer review of "Advances in Diagnosis, Pathological Mechanisms, Clinical Impact, and Future Therapeutic Perspectives in Tay–Sachs Disease"

_2035-8377, 2025, doi:10.3390/neurolint17070098_

Round 1
Reviewer 1 Report
Comments and Suggestions for Authors
My comments are attached.

Comments on the Quality of English Language
Rephrasing of the sentences is necessary to increase the impact of this review.
Author Response
Thank you sincerely for taking the time to review our article and for providing such constructive feedback. Your comments and suggestions helped us to refine and enhance the quality of our work. We deeply appreciate the care and attention you dedicated to evaluating our manuscript, as well as your expertise in identifying areas for improvement. Your efforts have strengthened the manuscript.
Major Points.
- According to your suggestion and examples, we have revised the English style of the whole manuscript.
- We have moved the order of topics in the text in parallel with the title according to your suggestion.
- We have modified the treatment section according to your suggestions.
Minor Points.
- According to your suggestions, several in vivo and in vitro studies from the literature have been added. Thus, some additional references have been included in the revised manuscript.
Reviewer 2 Report
Comments and Suggestions for Authors
Review
- The authors state in the abstract:
“Symptoms of Tay-Sachs typically appear between 20 three and six months of age and include delayed motor and cognitive development, fol- 21 lowed by progressive loss of function, blindness, paralysis, and death, usually before five 22 years of age.” This information is correct for the infantile phenotype of Tay-Sachs disease but does not apply to the juvenile and adult (late-onset) phenotypes. The authors should correct this statement.
- Due to pre-marital screening for carriers in many synagogues, the incidence of Tay-Sachs disease in Ashkenazi individuals has been reduced by around 95% in some areas of the world, including the United States, so it is incorrect for the authors to say that the disease occurs mostly in individuals of Ashkenazi Jewish heritage.
- To my knowledge, there is only 1 mouse study researching gemfibrozil for Tay-Sachs disease and an erratum has been published n this article, so the authors should clarify that this involves a mouse study (not human study).
Raha, S., Dutta, D., Paidi, R. K., & Pahan, K. (2024). Correction: Raha et al. Lipid-Lowering Drug Gemfibrozil Protects Mice from Tay-Sachs Disease via Peroxisome Proliferator-Activated Receptor α. Cells 2023, 12, 2791. Cells, 13(16), 1374. https://doi.org/10.3390/cells13161374
- Authors state:” Clinically, these two disorders are often indistinguishable and can only be reliably differentiated through enzymatic or genetic testing”.
There are actually some differences in the symptomology of these diseases, as we are learning through our more recent natural history studies during the past 20 years. The authors might say that the 2 diseases can look very similar, but there are some distinguishing characteristics that are becoming more clear through natural history studies.
- Please see the following studies on infantile and juvenile gangliosidoses:
Jarnes Utz, J. R., Kim, S., King, K., Ziegler, R., Schema, L., Redtree, E. S., & Whitley, C. B. (2017). Infantile gangliosidoses: Mapping a timeline of clinical changes. Molecular genetics and metabolism, 121(2), 170–179. https://doi.org/10.1016/j.ymgme.2017.04.011
King, K. E., Kim, S., Whitley, C. B., & Jarnes-Utz, J. R. (2020). The juvenile gangliosidoses: A timeline of clinical change. Molecular genetics and metabolism reports, 25, 100676. https://doi.org/10.1016/j.ymgmr.2020.100676
These natural history studies should be cited and mention of how the disease affects developmental milestones in children.
- Figure 1 refers to infant pictured as a male. Tay-Sachs disease is not typically thought of as gender specific. Please correct this figure.
- In the section on neuroinflammation, the authors should summarize findings from Utz, et al. (2014) and findings of inflammatory markers.
Utz, J. R., Crutcher, T., Schneider, J., Sorgen, P., & Whitley, C. B. (2015). Biomarkers of central nervous system inflammation in infantile and juvenile gangliosidoses. Molecular genetics and metabolism, 114(2), 274–280. https://doi.org/10.1016/j.ymgme.2014.11.015
- Authors state:
“It is important to highlight that TSD presents with a wide spectrum of neurological symptoms and signs, while systemic involvement is typically absent in these patients [88]. This contrasts with SD, where affected individuals not only exhibit degenerative neuro- ogical symptoms but also organomegaly, such as hepatosplenomegaly, which is not seen in TSD. “
Authors should clarify that patients with Sandhoff disease also experience a “wide spectrum” of neurological signs and symptoms. The way the manuscript is worded at this time, makes it sound like only Tay-Sachs patients experience a wide spectrum of symptoms.
- Authors state: “The gold standard for diagnosis is the measurement of enzyme activity in fibroblasts, chorionic villi or leukocytes”. The gold standard should also include molecular diagnostics.
- Cherry red spots are not always present at time of diagnosis. It is important to note that cherry red spots may develop later if they are not present at time of diagnosis.
- Authors should explain what the substance “propagermanium” is.
Author Response
Thank you so much for your enthusiastic feedback and for taking the time to review our article. We’re thrilled to hear your positive response and deeply appreciate your support in recommending it for publication. Your encouragement means a lot to us, and we’re grateful for your time and confidence in our work.
Comments.
- We have corrected the statement according to the reviewer´s suggestion.
- The paragraph has been modified to include the statement suggested by the reviewer.
- We have modified the paragraph related to the mouse model and the use of gemfibrozil for TSD according to the reviewer´s suggestions.
- We have modified the paragraphs related to symptomatology according to the reviewer´s suggestion to include recent natural history studies and how the disease affects developmental milestones in children.
- Figure 1 has been modified as suggested.
- We have included the findings of Utz and colleagues in the neuroinflammation section.
- The paragraph has been rewritten to be clarified, according to the reviewer´s suggestion.
- Molecular diagnostic has now been included as gold standard.
- The importance of cherry red spots at the time of diagnosis or later has been now stated.
- We have explained what propagermanium is.
We have tried to improve our manuscript with your considerations.
Round 2
Reviewer 1 Report
Comments and Suggestions for Authors
I have no comments
Author Response
Thank you very much for your interest in our work.
Kind regards,
Reviewer 2 Report
Comments and Suggestions for Authors
Please see the attached manuscript with my edits put into text boxes.
Throughout the paper, the authors were making too many statements in a conclusive way.
It is clear the authors have not worked directly with patients in clinical care setting and are not experienced in the natural history of GM2-gangliosidoses. This is not a bad thing. But the result of this is that I needed to make many corrections in order to make the paper more accurate.
I had to make many edits to try to make the paper more accurate.

Author Response
We have incorporated all the modifications recommended by the reviewer. We sincerely thank you for your interest in our manuscript, which has allowed us to present the most accurate depiction possible within the clinical context. Your input has been instrumental in ensuring that the manuscript adheres to the highest standards of quality and is of maximal relevance and interest to the readership.